# Position: Medical AI Neglects Real Treatment Outcomes

**Shiva Kaul** [1]    **Anjum Khurshid** [2]

## Abstract

Medical AI has rapidly improved its ability to perform diagnostic and prognostic tasks that lead to treatment decisions. But understanding of treatment itself is still inadequately trained and evaluated, using human opinions and syntheses (especially texts such as biomedical publications and clinical practice guidelines) rather than actual underlying data on treatment outcomes. This neglect seriously limits the potential of medical AI, and is already causing deficiencies in both frontier models and major benchmarks, as argued in this position paper. Real treatment outcomes, drawn from sources such as observational databases and randomized experiments, should be substantially incorporated into both training and evaluation. Improving these outcomes should be reemphasized as the downstream goal of all medical AI.

## 1. Introduction

In the 1990s, the field of medicine addressed two major impediments to quality care. The first problem was that too many medical decisions relied solely on the opinions of individual experts. Evidence-based medicine sought to base these decisions on the collective experience, and accumulated ground-truth data, of the broader medical system (Guyatt et al., 1992; Sackett et al., 1996). The second problem was that medical research, particularly clinical trials, had become myopically focused on surrogate (or intermediate) endpoints, such as serum concentrations of various analytes (Fleming & DeMets, 1996; Temple, 1999; Yudkin et al., 2011). Focusing on more clinically-relevant treatment outcomes, such as reductions in cardiovascular mortality, was a key thrust of patient-centered care.

In our opinion, similar problems currently afflict research in medical AI. The prevailing tendency is to focus on intermediate predictive problems, typically involving diagnosis or prognosis, without analysis of downstream improvements upon patient outcomes. Language models do not demonstrate a rich understanding of treatment effects, nor are they trained to develop it; instead, such causal reasoning is outsourced to preexisting works, which are relatively coarse and often mask both uncertainty and heterogeneity. During both training and evaluation of language models, human opinions are frequently treated as ground truth, even when better alternatives exist. This methodology, though common in other applications of AI, makes it challenging to achieve (or even recognize) performance exceeding that of any human expert — in other words, it contradicts not just the purpose of evidence-based medicine, but also renewed aspirations for artificial superintelligence (Morris et al., 2024).

Our position is that **medical AI currently neglects real treatment outcomes** [1]**, from training to evaluation, which hinders its purpose of improving these outcomes**. Our recommendation is to use more such data in medical AI research. This recommendation echoes those from the 1990s about incorporating ground-truth data into clinical practice. Our position is not just about realigning effort within medical AI research, but about expanding its long-term ambitions. Put simply, the long-term goal of medical AI should be to help write clinical practice guidelines and other authoritative syntheses, rather than to merely read and reference them.

Section 2 highlights the near absence of real treatment outcomes in the training of modern medical AI. Section 3 discusses why opinions and human-written syntheses, such as clinical guidelines and regulatory documents, have fundamental limitations as ground truth, during both inference and evaluation. Section 4 shows how established benchmarks focus primarily on ancillary or intermediate predictive tasks. It emphasizes the fact that treatment is the purpose of the medical system, and improving intermediate predictions doesn't always serve that purpose. We present our concrete recommendations in Section 5. Our position cuts against the grain of many research practices in medical AI, and is not without difficulties of its own. Section 6 discusses some reasonable alternative stances.

---

[1]Work performed at Department of Population Medicine prior to current affiliation. [2]Department of Population Medicine, Harvard Pilgrim Health Care Institute and Harvard Medical School. Correspondence to: Shiva Kaul <me@shivakaul.com>.

*Proceedings of the 43rd International Conference on Machine Learning*, Seoul, South Korea. PMLR 306, 2026. Copyright 2026 by the author(s).

---

[1]*Real treatment outcomes* are the actual outcomes of real patients observed in conjunction with their prior treatment.

| Model | Med-Gemini-* (Yang et al., 2024) | Med-Gemini-[L,M] (Saab et al., 2024) | AMIE (Tu et al., 2025) | Med-PaLM 2 (Singhal et al., 2025) | Med-PaLM (Singhal et al., 2023) | Baichuan-M3 (Baichuan-M3 Team, 2026) | OpenBioLLM (Pal & Sankarasubbu, 2024) | Me-LLaMA (Xie et al., 2025) | MEDITRON (Chen et al., 2023) | Med42-v2 (Christophe et al., 2025) | MedGemma (Sellergren et al., 2025) | TxAgent (Gao et al., 2025) | HealthGPT-Pro-8B (Lin et al., 2025) | LLaVA-Med (Li et al., 2023) | BioMistral (Labrak et al., 2024) | MediPhi (Corbeil et al., 2025) | Curiosity-L (Waxler et al., 2025) | Mamba-CLMBR (Wornow et al., 2025) | EHRMamba (Fallahpour et al., 2024) | MOTOR-T (Steinberg et al., 2024) |
|---|---|---|---|---|---|---|---|---|---|---|---|---|---|---|---|---|---|---|---|---|
| **Size** | — | — | — | — | 540B | 235B | 70B | 70B | 70B | 70B | 27B | 8B | 8B | 7B | 7B | 3.8B | 1B | 121M | 130M | 143M |
| **Web Crawl** | √ | √ | √ | √ | √ | √ | √ | √ | √ | √ | √ | √ | √ | √ | √ | √ | | | | |
| **QA Training Splits** | √ | √ | √ | √ | √ | √ | ✓ | √ | √ | √ | √ | | √ | √ | √ | √ | | | | |
| **Publications** | √ | √ | ✓ | ✓ | ✓ | √ | ✓ | √ | √ | √ | √ | √ | √ | √ | √ | √ | | | | |
| **Clinical Trials** | ✓ | ✓ | ✓ | ✓ | ✓ | ✓ | ✓ | √ | √ | ✓ | √ | | | √ | √ | | | | | |
| **Guidelines** | ✓ | ✓ | √ | ✓ | ✓ | √ | ✓ | √ | √ | ✓ | √ | | | | ✓ | √ | | | | |
| **Real Patient Data** | √ | √ | √ | | | | ✓ | √ | | | √ | √ | √ | √ | | | √ | √ | √ | √ |
| **Real Treatment Outcomes** | * | | | | | | | | | | | | | | | | √ | √ | √ | √ |

\* Yang et al. (2024) use genomic and outcome data, but not treatment data, from the UK BioBank.

*Table 1.* The different kinds of data used to train modern language models (and a few EHR models) for general medical purposes. (See Wornow et al. (2023) for similar information on models trained before 2023). A large checkmark √ denotes explicitly-mentioned inclusion of the data in the model's training set; a small one ✓ means implicit inclusion through e.g. an extensive web crawl retained in the weights of a large model. "Patient data" includes imaging, notes, diagnoses, etc. of real patients, beyond cases presented in medical literature. "Real treatment outcomes" refer to treatment records along with clinical outcomes (e.g. discharge, symptom resolution, death) of real patients. The largest models train on a panoply of data, including real patient data. However, only the smaller EHR foundation models train on real treatment outcomes. *Caveat*: the boundaries between some training and inference sets are unclear.

**An olive branch** This paper unavoidably critiques many recent works in medical AI. We believe these works made valuable advances, usually in a very short period of time, which we do not undermine. Rather, this paper charts a course to maintain their exciting pace of progress.

## 2. Doctors Train From Clinical Observation of Treatment; AI Does Not

To practice with a board certification in the United States, a physician must complete a clinical residency which typically lasts as long as their academic medical schooling. In this setting, they directly observe the consequences of interventions upon patients, and thereby develop their clinical competence. Observing real treatment outcomes is a crucial aspect of physician training, but it is essentially absent in the training of modern medical AI. Table 1 characterizes the training corpora of such models. Due to space constraints, and the rapid pace of model development, we exclude models trained prior to 2023. In this table, and throughout the paper, we focus on foundation models (primarily LLMs) that are trained for general medical use by a broad audience, excluding models trained for highly specific tasks.

We observe that most medical AI training is conducted on textual material such as publications and clinical guidelines. This is unsurprising, since most of these models derive from base language models trained primarily from the web.

Most models are fine-tuned on the training splits of medical question-answering datasets. Many are trained on real patient data, especially clinical notes and radiological images. Bedi et al. (2024) recently observed that, despite such training, most subsequent evaluations of medical AI are based on synthetic cases rather than real patient data.

For training, we observe the major watershed is not between real and synthetic data, but between the inclusion or exclusion of real treatment outcomes. To the best of our knowledge, the only major class of foundation models which include such data are those trained solely from electronic health records. Such EHR models do not operate on language, but rather sequences of medical events. Treatment outcomes are inherently longitudinal — for each patient, their treatment must be paired with their subsequent outcome — and such data are the purview of observational medical databases.

## 3. The Use of Opinions and Syntheses as Ground Truth About Treatments

### 3.1. Medication Labels

Medication labels, such as FDA product labels, are government-sanctioned communications of approved uses, risks, and instructions. They form the basis for lawful product marketing. These labels typically range from 15 to 100 pages, compile the most crucial information about the

**Case**: 28-year-old female / Medical history: Non-vitamin B12 responsive methylmalonic acidemia (MMA) / Presenting with: Chronic pancreatitis (paraduodenal subtype) / Clinical severity: M-ANNHEIM Ib

*Clinical Presentation*: 9 hospital admissions over one year for acute pancreatitis / No family history of pancreatitis / No alcohol use / Primary nutrition: Protein-restricted metabolic formula with canola oil / Normal stool fats

*Diagnostic Findings*: Paraduodenal pancreatitis confirmed on T1-weighted sequence of an MRI pancreas with and without contrast / Met American Pancreatic Association criteria for chronic pancreatitis based on clinical history and CT imaging / Average baseline lipase level: 72 U/L (reference range 9-82 U/L) / Random triglyceride measurements during acute episodes did not support triglyceride-induced pancreatitis / MMA prevented true fasting triglyceride measurement

*Genetic Investigation*: Broad workup including genetic causes of recurrent pancreatitis / Identified: Heterozygous R668C (c.2002C>T) variant of unknown significance in CFTR gene

*Declined Testing*: Patient declined sweat chloride test / Patient declined nasal potential difference testing / Reason for declining: Perceived discomfort

**Question:** Characterize the utility of ivacaftor for resolving this patient's pancreatitis.

**(A)** Contraindicated for adverse effects.   **(B)** Effective, with high certainty.

**(C)** Effective, with low certainty.   **(D)** Ineffective.

*Figure 1.* A mislabeling that occurs when regulatory documents and LLMs are used to (synthetically) validate labels for benchmarks, in a manner reminiscent of Palepu et al. (2025) and Gao et al. (2025). This question derives from a real, published case report in which ivacaftor resolved a patient's pancreatitis (Tang et al., 2022). The correct answer is (C); the supporting evidence for this answer is highlighted green, and their connection is explained in the main text. Unfortunately, the incorrect answer (D) passes both of the automated correctness checks of Palepu et al. (2025), which are based on the FDA product label for ivacaftor. The fundamental problem is that regulatory documents, such as drug labels, are not intended to provide comprehensive ground truth for treatment outcomes. Note that the RxQA dataset is additionally verified by pharmacists, and for clarity, whereas TreatmentPC is not.

medication in question, are free to access, and are published in the public domain. As such, they are convenient source material for NLP (Li et al., 2013) and model training (ValizadehAslani et al., 2023). Recently, Palepu et al. (2025) and Gao et al. (2025) used medication labels to produce question-answer benchmarks named RxQA and TreatmentPC, respectively. The latter's methodology underpinned the CURE-Bench Challenge at NeurIPS 2025. In these datasets, each question presents a patient case study along with multiple-choice answers. They concern different aspects of treatment planning. The questions are automatically generated and validated in a way that is based primarily on medication labels. (RxQa is additionally verified by pharmacists, but TreatmentPC is not).

The purpose of medication labels is to legally constrain the marketing activities of manufacturers. Multiple court opinions have determined that medication labels do not represent a standard of care (Beck, 2017). Importantly, they do not discuss *off-label use*, which is a fundamental aspect of medical practice. In some fields, off-label use constitutes 30-80% of all medication use (Allen et al., 2018).

To demonstrate the perils of treating medication labels as ground truth, we demonstrate a bypass of both of the automated correctness verification measures used in RxQA. [2] The prompts for these verifications are displayed in Figure 4,

Appendix A.2.1. In the first, the language model is given the medication label and the generated example, and is asked whether the candidate correct answer is, in fact, correct. The second verification presents the same inputs, but asks if the other answers are, in fact, incorrect. We present an example, shown in Figure 1, where the incorrect answer (D) passes both verifications. Figures 5 and 6 show Gemini 1.5 Flash, which was used by Palepu et al. (2025), performing these erroneous verifications. (Our results also hold for Gemini 3.5 Flash and Gemini 3.1 Pro Preview, so we will simply refer to the model as Gemini.)

**Background**  Our example in Figure 1 is based on the real, published case report of Tang et al. (2022), who encountered a patient suffering from idiopathic pancreatitis. After finding a heterozygous variant in the patient's CFTR gene, they administered ivacaftor, which completely resolved the patient's pancreatitis. Ivacaftor belongs to a class of drugs called CFTR modulators which target defective CFTR proteins. The motivation for the treament was an emerging (low-certainty) body of evidence which ties a large fraction of idiopathic pancreatitis (perhaps 30-40%) to CFTR gene alterations, or even environmental factors affecting CFTR expression (Cohn et al., 1998; Audrézet et al., 2002; Phadke & Sellers, 2022; Hegyi et al., 2016). Such irregularities and dysfunction may not be severe enough to earn a full diagnosis of cystic fibrosis — that is, a threshold value of $\geq 60$ mmol/L on a sweat chloride test — which would then classify the use of ivacaftor as on-label.

_____________

[2] RxQA has an additional clarity check which we don't involve, and arguably promotes simplistic questions.

**Ablations** The immediate counterargument to our experiment is: a case report is just a single datum, and the supporting evidence is weak. Perhaps Gemini is simply being logical and evidence-based in its continued refutation of ivacaftor's potential effectiveness. To examine this possibility, we repeated the experiment, but additionally attached the case report. Per Figure 8, Gemini finds the case report convincing enough to correctly answer (C). Is it possible that the mere inclusion of a case report, generally discussing ivacaftor's effectiveness, was sufficient to overwhelm Gemini's clinical reasoning? To check this, we synthesized a patient who would almost certainly not benefit from ivacaftor, as their pancreatitis seems caused by alcohol use (Figure 7). After attaching the case report and asking about this synthetic patient (Figure 9), Gemini retains its clinical reasoning capabilities and correctly answers (D). The problem lies not in Gemini's abstract reasoning capabilities, but in the text it was told to treat as ground truth.

### 3.2. Clinical Practice Guidelines

Clinical practice guidelines are extensive consensus statements published by professional societies, governmental health bodies, and international organizations. Within a specific area of practice, they discuss many aspects of care, including diagnostic criteria and risk stratification. They usually include decision trees for recommended treatment pathways. They typically take years to write and are sometimes hundreds of pages long. As seen in Table 1, guidelines feature in the training corpora of many medical language models. At inference time, some models are designed to specifically, or even exclusively, cite guidelines. For example, the Mx Agent of AMIE generates treatment plans by (1) searching solely for clinical practice guidelines, and (2) citing those guidelines via in-context retrieval (Palepu et al., 2025). Services such as OpenEvidence and ChatGPT for Healthcare are designed to cite authoritative documents such as guidelines. AMEGA is a recent, rubric-based benchmark which measures adherence to guidelines (Fast et al., 2024).

The purpose of clinical practice guidelines is to establish generic professional standards. However, they are not binding either professionally or legally, even when determining a minimum standard of care. The American Law Institute recently clarified the legal role of guidelines in their restatement of torts, which revised their widely held legal standard for medical negligence (Aaron et al., 2025; Peters Jr, 2023). Although adhering to guidelines may exculpate a physician from malpractice claims, deviating from them does *not*, in of itself, indicate malpractice. This asymmetry means grounding an answer in guidelines can lead to reasoning that is merely defensible rather than accurate.

Guidelines do not describe optimal or personalized treatment. When basing answers primarily (or solely) upon guidelines, the following deficiencies can arise:

- Oversimplification: guidelines are written for, and by, people. This makes their treatment plans highly constrained in terms of decision-tree complexity. They do not gracefully handle stochasticity, express backtracking, or address a long tail of nonstandard circumstances.
- Staleness: guideline creation is a formal process which convenes many experts, so these documents are updated very infrequently. Real clinicians are expected to routinely resolve conflicts among guidelines, new evidence, and their own experience.

We present two concrete examples of these deficiencies. The first example is presented in Figure 2. It is based on the real, published case report of Ha et al. (2021), in which a physician successfully treated a cardiac patient by reducing their pill burden. This case illustrates the phenomenon of *prescribing cascade*, in which symptoms are mistakenly thought to arise from a disease process, when they are actually the side effects of a previously administered drug (Rochon & Gurwitz, 1997; Brath et al., 2018). Guidelines do not typically provide alternate recommendations for nonadherent patients. Guidelines also don't address prescribing cascade, since it is a path-dependent phenomenon whose resolution involves backtracking. As a consequence, the explicitly guideline-directed answer provided by OpenEvidence did not recognize the prescribing cascade: it increased, not decreased, the dose of furosemide. Furthermore, it did not attempt to reduce the patient's pill burden. As seen in Figures 13 and 14 of the Appendix, ChatGPT for Clinicians and ChatGPT for Healthcare similarly failed to detect the prescribing cascade. They did not recommend reducing furosemide, although they fared slightly better by at least recognizing the importance of medication adherence.

Figure 15, in Appendix A.3, presents the second example, as well as the clinical reasoning and treatment plan provided by OpenEvidence. This example is based on a published report of how OpenEvidence (retrospectively) planned treatment for a real patient at the Mayo Clinic (Hurt et al., 2025). The focus of our scrutiny is the reasoning and use of evidence, rather than the correctness of the final treatment plan. The answer ignores important information — in particular, precision lab measurements — in deference to older 2018 guidelines, which focus on a more widely-available proxy measure. Furthermore, the answer does not incorporate information from other (newer) guidelines and sources, or even hint that those sources may present different reasoning about lab measurements.

### 3.3. Clinician Opinion (And Aggregates Thereof)

The limitations of individual human judgment have been studied extensively in psychology (Meehl, 1954; Tversky & Kahneman, 1974). These motivated the use of evidence-

An 87-year-old female with a documented medical history of atrial fibrillation, chronic heart failure with preserved ejection fraction (HFpEF), hypertension, hypokalemia, vitamin D deficiency, and glaucoma is taking the following medications:

Aspirin 81 mg, once daily / Diltiazem 240 mg extended-release, once daily / Furosemide 40 mg, once daily / Metoprolol tartrate 50 mg, twice daily / Potassium chloride 20 mEq, once daily

The patient is experiencing moderate pedal edema and has an unstable gait. Medication non-adherence is an ongoing issue. Additionally, a recent echocardiogram indicates that the patient's Left Ventricular Ejection Fraction (LVEF) has worsened to less than 40%. **What are the next steps of treatment?**

- Diltiazem: consider sacubitril-valsartan for further "optimization of guideline-directed medical therapy."
- Furosemide: increase dose to address volume overload and pedal edema.
- Metoprolol tartrate: consider succinate for hypertension management - no consideration for pill burden.
- *Added*: SGLT2 inhibitors and spironolactone.

- Diltiazem: discontinued. Replaced by losartam due to HFrEF, edema.
- Furosemide: reduce dose because of cascade from diltiazem to edema.
- Metoprolol tartrate: switch to succinate to reduce pill burden (once daily).
- *No additions*

*Figure 2.* An example of how formulaically grounding answers in guidelines can harm clinical reasoning. The question is based on the published case report of Ha et al. (2021), which involved a prescribing cascade: the patient's potassium deficiency (hypokalemia) was exacerbated by the furosemide, which was being given to address edema, a side effect of diltiazem. The physician's treatment plan is on the right in light green. It recognized the cascade and successfully treated both the patient's heart condition and edema. It also reduced pill burden in light of the patient's nonadherence. On the left is a summary of OpenEvidence's treatment plan. Its full response is available at this anonymized link and also in Figure 12, Appendix A.3. It explicitly describes its answer as "guideline-directed." Indeed, it maintains the focus of the guidelines, correctly addressing HFrEF. However, it ignores the patient-specific considerations in yellow. Furthermore, it doesn't reason about the prescribing cascade: it elects to increase, not decrease, furosemide dosage.

based methodology in medicine (Cochrane, 1972), where the fundamental problem of causal inference makes them especially acute. Since clinicians don't see the counterfactual outcomes of patients they treat, their ability to independently accumulate expertise about treatment is limited.

Meanwhile, clinician opinion is routinely used as ground truth in medical AI benchmarks. Clinicians are queried "how does the answer relate to the consensus in the scientific and clinical community?" (Singhal et al., 2023) or "which answer better reflects the current consensus of the scientific and clinical community?" (Singhal et al., 2025). This clinician involvement is not necessarily bad. Medical AI needs to be judged for qualities such as empathy, adherence to social norms, and ethical alignment — topics in which people are the ultimate arbiters.

However, measuring agreement with groups of clinicians is not a general replacement for ground truth validation. Agreement has no strict relationship with the truth. A majority consensus usually has lower average error than the average individual clinician (by convexity). But consensus can be expensive to establish, and agreement may not actually reflect consensus. This dilemma echoes related concerns about annotation in natural language processing (Plank, 2022). A disappointing aspect of agreement as an evaluation metric is that it halts our ambitions to understand what humans do not already know.

These issues come to the fore in HealthBench (Arora et al., 2025). In this benchmark, language models are presented with a health-related question or conversation. Their answers are judged according to criteria specified by clinicians. For the present discussion, the most relevant criterion is "this answer contains no factually incorrect information", but there are many different axes of evaluation. At evaluation time, GPT 4.1 grades whether an answer meets the prespecified criteria. HealthBench is notable because it offers a meta-evaluation of how well GPT 4.1 performs as a grader compared to clinicians. In this evaluation, clinicians provide 60,896 true/false labelings of whether an answer to a question meets a specific criterion. In this crucial aspect of rubric benchmarking, Arora et al. (2025) conclude that GPT 4.1 achieves "clinician-level agreement."

Let us examine what this means. The dataset involves no ground truth outside of clinician opinion. Because of the sparsity of the labeling, clinician consensus is very tenuous, or does not exist, for the vast majority of the examples. Concretely, 55,546 (91.2%) of the labels originate from 27,773 examples which were labeled by exactly 2 clinicians. (Whether the criterion is relevant to the question *is* previously determined by a consensus, but whether any given answer satisfies it is not). Thus, agreement cannot be reasonably assessed on a per-example basis. Instead, clinicians (and GPT 4.1) are scored on how much they agree with (other) clinicians across all examples.

As a consequence of this design, it is possible for GPT 4.1 to have high agreement and low accuracy at the same time. In particular, let us examine the 5,105 HealthBench meta-examples about whether an answer was factually accurate. "Clinician-level agreement" still holds on this subset. All 5,105 such examples were labeled by GPT 4.1 and just 2 clinicians. Their labels group as follows.

| GPT 4.1 | Clinicians | Count |
|---------|-----------|-------|
| Yes | Both Yes | 3175 |
| | Disagree | 1050 |
| | Both No | 246 |
| No | Both Yes | 221 |
| | Disagree | 208 |
| | Both No | 205 |

Consider the following possible ground-truth labeling: when GPT 4.1 and the clinicians all agree (in $3175 + 205 = 3380$ examples, i.e. 66.2%) then set the label to match everyone. In the remaining examples, set the label to the opposite of GPT's. This would make the clinician answers 87.6% accurate compared to GPT's 66.2%. By the opposite construction, it is also possible that GPT 4.1 is 100% accurate and clinician answers are only 78.5% accurate. Thus, in this context, agreement is just a marginal guarantee of conformity unmoored from the truth.

### 3.3.1. WHAT IS ELICITED FROM CLINICIANS?

The theoretical hope of querying clinical experts is that they are the ultimate aggregation of all medical knowledge: they have both hands-on experience and knowledge of guidelines and other authoritative documents. In principle, asking clinicians to label examples should give higher quality than any of these underlying sources. However, in reality, it may be the case that, when faced with an abstract grading task, experts merely defer to such documents rather than conveying the full breadth of their expertise.

To investigate this, we examine HealthBench Professional (Hicks et al., 2026). Like HealthBench, this benchmark is based completely on clinician input, with no incorporation of ground-truth clinical data. There are 525 examples of conversations (initiated by medical professionals) and sample answers. Across these examples, there are 1135 total clinician-established criteria for grading answers. In our analysis (detailed in Appendix A.4) we find that, among the criteria relating to clinical facts or decisions, approximately 86% are directly grounded in a passage from a guideline or another such authoritative document.

This is substantially higher than clinician guideline concordance in practice. Large-scale studies, averaging across different specialties, generally estimate real-world guideline concordance to hover around only 55% to 60% (McG-

lynn et al., 2003; Runciman et al., 2012). The reasons for guideline discordance vary: some of this is likely due to outdated training or lack of knowledge. However, a recent study (Finkelstein et al., 2022) found that physicians may be less guideline-concordant when treating their own family members, suggesting that they may take a more nuanced, personalized approach in sensitive circumstances. Regardless of the underlying reason, this statistical discrepancy indicates that clinician benchmark input and actual clinician behavior may differ.

## 4. Treatment Is the Focus of Medicine, But Not of AI Evaluations

The purpose of the medical system is to improve outcomes through treatment. This basic intuition is reflected in both the law and health economics. Diagnosis and ancillary tasks may be carried out by different kinds of professionals, but the prescriptive authority to initiate significant treatment is reserved for licensed physicians. Since the passage of the Affordable Care Act (2010) and MACRA (2015), value-based care has taken root across US healthcare policy, increasingly tying provider reimbursement to treatment outcomes. Despite the centrality of treatment, most established AI evaluations focus on intermediate predictive tasks. We now discuss the extent to which they dominate benchmarks, and why it is important to elucidate their relationship with improved treatment outcomes.

### 4.1. Prevalent AI Benchmarks Don't Assess Understanding of Treatment Outcomes

Until recently, the primary benchmarks in medical AI were collections of multiple-choice questions. Most of these focused on general biomedical knowledge, such as PubMedQA (Jin et al., 2019) and MMLU clinical topics (Hendrycks et al., 2021). Understanding of treatment outcomes was measured by performance on datasets of medical exam (e.g. USMLE) questions. However, these initial datasets, such as MedQA (Jin et al., 2021) and MedMCQA (Pal et al., 2022), have now become saturated (Saab et al., 2024). Furthermore, their limitations in measuring actual clinical understanding are recognized (Raji et al., 2025; Alaa et al., 2025). Newer QA datasets require more difficult free-form answers — notably, ones derived from JAMA Clinical Challenge (Chen et al., 2025a; Chang & Fontanarosa, 2011). Newer benchmarking efforts evaluate LLMs more comprehensively along more clinically important axes. Perhaps the most significant effort in this direction is MedHELM (Bedi et al., 2026), which aggregates 37 datasets in view of 121 clinically important tasks.

Though these new benchmarks offer many substantial improvements, they (still) neglect to meaningfully evaluate understanding of treatment outcomes. By this, we refer to

questions which are based on realistic cases, and ask for a free-form treatment recommendation (i.e. "what are the next steps of treatment?") or more quantitative estimates akin to regression ("what is the effect of this treatment?"), ranking ("what is the best treatment?") or classification ("is this treatment effective?").

Relatively few entries in JAMA Clinical Challenge fit into these categories. This repository consists of 1700+ questions, of which most (1524) are in the dataset of Chen et al. (2025a). By our calculations, of the 1277 questions with public information, 711 (55.6%) ask for a diagnosis, 36 (2.8%) are about unknown or assorted topics, and 530 (41.5%) ask for next steps of treatment. Of the "next step" questions, approximately 232 (18.1% of the verifiable questions) actually involve some assessment or modification of treatment; the others are answered just by requests for further information via imaging, biopsies, and so forth.

Of the 31 datasets in MedHELM, only two nominally pertain to treatment planning: MTSamples and MEDEC (Ben Abacha et al., 2025), which are both clinical note datasets. In MTSamples, the model is presented with the initial part of a note, and is prompted to generate a treatment plan. Its output is then compared to a later part of the note, often a recap of a surgical operation. Recording patient interactions is only tangentially related to providing ground truth about treatment outcomes. In MEDEC, the goal is to take a note and identify which, if any, of the sentences contain an error. As seen in Figure 3 of Ben Abacha et al. (2025), only about 10% of these errors relate to treatment. Most of the dataset is generated by using MedQA as a source of ground truth for scenarios. Retrospective correction of errors — of which some are transcription flaws, and others are iatrogenesis — is not the same task as prospective treatment planning.

Aside from these conceptually-oriented benchmarks, there are numerous clinical evaluations of LLMs in the medical literature. These were the subject of a recent systematic review by Chen et al. (2026), which found 789 out of 4609 clinical evaluations of LLMs (17.1%) nominally involve treatment planning or recommendation. We further scrutinize these 789 studies, checking (1) if they actually focused on some aspect of treatment itself, rather than diagnosis, prognosis, or other intermediate tasks, (2) whether they could be reused as benchmarks, as opposed to being one-time manual assessments of answers, and (3) whether real treatment outcomes were used as ground truth, as opposed to guidelines or subjective expert opinions. (Full details are in Appendix A.5.) We find that just $180/4609 \approx 3.9\%$ of the studies are potentially-reusable benchmarks focused on treatment. Furthermore, the vast majority of ground truth is unmoored from patient outcomes, with $702/789 \approx 88.9\%$ of the subset relying on guidelines or human opinion. There are only $16/4609$ reusable benchmarks about treatment

which involve real outcomes. To summarize, even nominally treatment-related evaluations do not assess a causal understanding of treatment, and the use of textual syntheses or opinions as ground truth is pervasive.

## 4.2. Therapeutic Development is Not Patient-Centered Treatment Understanding

Multiple benchmarks assess how well models predict clinical trial outcomes. These include TOP (Fu et al., 2022), TrialBench (Chen et al., 2025b) and CTO (Gao et al., 2024). Clinical trials offer crucial ground-truth data on treatment outcomes. However, these benchmarks are oriented towards reducing the cost of therapeutic development, and only partially align with the goal of understanding treatment effects. They lack a focus on comparative effectiveness — determining when one treatment is superior to another — which is one of our main concerns.

In therapeutics research, a clinical trial is deemed successful if it meets statistical endpoints and proceeds to the next phase. In Phases 1 and 2, the decision to advance is made by the sponsor, frequently driven by portfolio strategy rather than clinical utility alone. By Phase 3, success is defined by regulatory approval, but the FDA is statutorily barred from considering cost and generally does not mandate evidence of superiority to existing treatments (Gottlieb, 2011).

As such, commercial and operational factors can influence the outcome definitions in these benchmarks. For example, TrialBench marks negative outcomes for insufficient patient enrollment. Also, CTO infers missing outcomes from stock prices and similarly marks trials terminated for administrative reasons as failures.

## 4.3. Connecting Intermediate Predictions to Outcomes

In medicine, automating some kinds of intermediate and ancillary tasks (such as ambient clinical documentation, automated billing, or triage scheduling) demonstrably reduces administrative burden and provider burnout (Afshar et al., 2025).However, the connection between intermediate predictive tasks and patient outcomes is not always clear.

Perhaps the most well-known examples of such surprises arise in preventive screening. Breast cancer (mammography), prostate cancer (prostate-specific antigen), and lung cancer (low dose CT) screening tests have all been found to be less worthwhile than anticipated, following the completion of randomized trials (Miller et al., 2014; Andriole et al., 2009; Marcus et al., 2000). Screening, early diagnosis, and risk stratification are counterproductive when they find problems to medicalize rather than genuine opportunities to intervene and improve outcomes. For example, in colonoscopy, using computer vision to find the maximal number of polyps or lesions isn't necessarily ideal. Improving outcomes re-

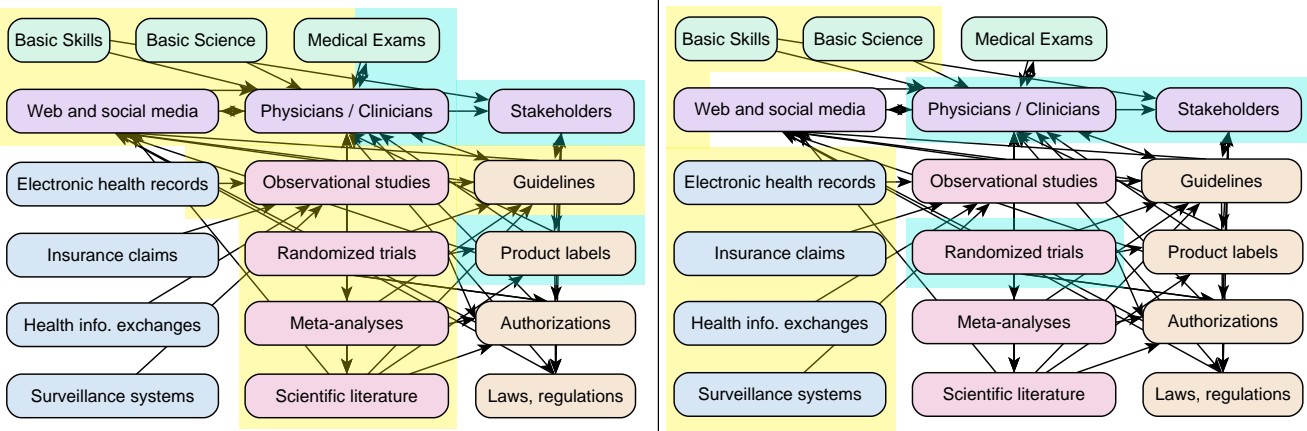

*Figure 3.* The convoluted relationships among different kinds of medical data — and two different ways that AI can approach such data. On the **left** is the status quo; yellow highlights data used for training, and teal highlights data used for evaluation. As seen in Table 1, treatment outcomes from observational databases are essentially ignored: the partial shading denotes limited usage. Training is based primarily on human opinions and syntheses. Evaluation is performed on formally separate splits of such data. However, due to the close interconnections among such data sources, leakage is a serious concern. On the **right** is the cleaner system advocated by this paper. It treats the source nodes in this graph as ground truth. Training is based primarily on observational databases. Randomized trials are used to evaluate understanding of treatment effects. Approaches like full conformal prediction allow learning from trials during uncertainty quantification (Angelopoulos et al., 2024). Human opinions are used to evaluate intermediate predictions, alignment, etc. Overall, this approach reduces leakage. The ensuing models can be used to inform research, regulations and professional standards.

quires finding polyps which are not so slow-growing to be harmless within the course of a lifetime, but also not so aggressive that they will kill the patient regardless (Hassan et al., 2023). Harm can occur by subsequent risks in the medical system (i.e. overdiagnosis (Welch & Black, 2010)) or the data acquisition itself (for example, perforation in colonoscopy). In these scenarios, optimizing for raw diagnostic sensitivity rather than patient-centered benefit, without accounting for potential side effects or system-level constraints, decouples the intermediate prediction from the ultimate outcome.

These difficulties stem from fundamental misalignments between the predictive task and downstream outcomes. But even when those are conceptually aligned, more subtle problems can arise from the way model performance is quantified. Most benchmarks report some kind of average accuracy or win rate (Liang et al., 2023). However, the key property which enables predictions to be used safely for downstream decisions is calibration, not accuracy (Noarov & Roth, 2024). Unfortunately, calibration is challenging to both achieve and verify. Recent works have examined less stringent conditions which focus on specific kinds of downstream decisions (Zhao et al., 2021; Rothblum & Yona, 2023; Noarov et al., 2025).

## 5. Recommendations (Call to Action)

**Training** should be based more on longitudinal patient records (e.g. electronic health records and insurance claims), and less on textual syntheses such as clinical practice guide-

lines, medical publications and related literature. Longitudinal data pipelines are common in the real-world evidence (RWE) industry, which rapidly extracts causal insights from large-scale databases (Jensen et al., 2012; Sherman et al., 2016). Large extracts of such datasets are now being released, at no cost and without excessive encumbrance, to the research community. A major example is the CRITICAL dataset (Luo et al., 2024). This dataset involves approximately 400,000 critical care patients, including longitudinal inpatient and outpatient data from before, during, and after their ICU admission. Software for processing this dataset into standardized machine learning tasks is available (Luo & Li, 2025).

**Evaluation** should be based more on randomized experiments, where applicable, and less on regulatory documents, clinical practice guidelines, and clinician opinion. By this, we mean predicting the results of previously conducted experiments; having AI participate in new experiments may present ethical and logistical challenges. New benchmarks can be formulated as target trial emulations (Hernán & Robins, 2016; Hernán et al., 2022). When creating these, pipelines from trial outcome benchmarks can be reused (Chen et al., 2025b). When experimental data are not available, held-out observational data can be used to generate target causal estimates. Ground-truth quantitative outcome data, without additional interpretation, can be extracted from scientific publications. (However, care should be taken to avoid the leakage issue illustrated in Figure 3).

**Intermediate predictive tasks** should, of course, still be

vigorously pursued. Their contribution to improving treatment outcomes should be quantified, at least roughly. Ideally, this could occur by analyzing deployments of predictive models as causal interventions (Joshi et al., 2025) while reporting outcome-oriented metrics (Tierney et al., 2024). At earlier stages, translating performance improvements into more patient-oriented metrics, as done in health economics, would bring a mature emphasis on real-world priorities (Rossi et al., 2022; van Leeuwen et al., 2021).

For example, consider ambient AI summarization of patient conversations. During development, such models are benchmarked on NLP-style metrics such as Word Error Rate (WER) or Mean Text Recurrence (MTR), which are appropriate for algorithmic research. A complementary layer of evaluation connects these metrics to clinician performance (Moramarco et al., 2022) and, in turn, clinical experience and practitioner well-being (Afshar et al., 2025). This burden does not fall on algorithmic developers, but on researchers focusing on clinical evaluation and deployment.

## 6. Alternative Views

*"Training AI on longitudinal records is a violation of patient privacy."* It should be noted that medical AI models already train upon large quantities of sensitive, individual-level patient data (see Table 1), while respecting prevailing standards in biomedicine. Treatment outcomes are not inherently more sensitive than such data. Indeed, treatment effects are most naturally inferred at a group level. Thus, techniques such as differential privacy (Dwork, 2006) can be used to ethically learn about treatment effects.

*"Training pipelines for text are more mature than those for medical records."* Indeed, observational medical databases can be fragmented and difficult to access. By law, electronic health records must use standard formats (Bender & Sartipi, 2013) and vocabularies, and must be accessible via API (Phelan et al., 2024). However, these provisions ensure only syntactic, not semantic, interoperability (Dolin & Alschuler, 2011). We concede that most observational medical databases are not ready for large-scale AI training. Recent developments in AI depended on major concurrent investments in training data pipelines (Brown et al., 2020; Gao et al., 2020; Ratner et al., 2017). Medical AI seems to need the same (Khurshid & Sarkar, 2025).

*"Publications and other syntheses comprise the majority of readily-available medical training data."* Our recommendation is not extreme and absolute. We recommend a gradual shift to real treatment outcomes; we acknowledge this may need to occur over the course of years.

*"Even when they are available, randomized controlled trials have flaws as ground truth."* Low power, blinding violations, and dropouts can harm their internal validity (Deaton & Cartwright, 2018). Highly-artificial inclusion criteria can limit their external validity (Rothwell, 2005). Fortunately, these design aspects of the trial are features that can be used to predict its outcome. Thus, if a trial's design biases its outcome, then that bias can be disentangled from the true underlying effect (Kaul & Gordon, 2025). Another issue is that RCTs elucidate average, rather than individual, treatment effects. Individual patient data from RCTs can be used to make inferences about subgroups or (under stronger assumptions) individuals themselves (Pearl, 2015). Finally, RCTs are not the only kind of experiments which can serve as ground truth (Guyatt et al., 1986; Liang & Recht, 2025).

*"Conformance with guidelines and clinician consensus is a practical requirement for any deployable medical AI."* Our recommendation is to improve this status quo. Even leaving AI aside, medical evidence is rife with uncertainty and controversy (Shaneyfelt et al., 1999; Prasad et al., 2013; Kent & Ioannidis, 2026). Ideally, AI should help resolve these challenges, not just propagate them.

*"Focusing primarily on treatment outcomes risks devaluing intermediate tasks, which represent the most successful and practical applications of medical AI to date."* Placing emphasis on treatment outcomes does not necessarily diminish the value of intermediate tasks. This dynamic is familiar to the medical community: the historical shift toward evidence-based, patient-centered outcomes (Guyatt et al., 1992; Sackett et al., 1996) did not imperil the in-silico or in-vitro biomedical investigation of molecules and analytes. In the same way, training medical AI to better understand treatment will likely asssist in solving existing predictive tasks. For example, in ambient AI scribing, anticipating likely treatments helps a clinician ask for the vital patient information needed to complete a high-fidelity clinical note. This, in turn, helps downstream providers manage treatment more effectively. Viewing these tasks as mutually reinforcing and servicing the same ultimate goal enables collaboration, not antagonism, among different research agendas.

## 7. Conclusion

Making predictions by using published text as source material, and humans as ground truth, is a pattern adopted from other AI. It embodies neither the customs nor the ambitions of medicine. The best long-term plan for medical AI is to reemphasize what truly matters: real treatment outcomes.

**Acknowledgements** The authors thank Richard Platt and Geoffrey J. Gordon for their valuable comments and feedback.

**Reproducibility** Code and data supporting our analyses are available at `https://shivakaul.com/x/icml26`.

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

# A. Appendix

## A.1. Related Work

**Emphasizing outcomes in medical AI**   The recent paper of Joshi et al. (2025) also calls for greater emphasis on clinical outcomes in medical AI. However, their work is very different than ours. They recognize that deployments of predictive AI are interventions, and propose to analyze them from a causal perspective. They discuss, for example, the pitfalls of deploying a model which predicts heart disease from inexpensive ECGs, when its training was conducted on a smaller, biased population with expensive ground truth labels. To catch problems such as this distribution shift, they present a prospective (and continuing) validation framework using silent trials or off-policy evaluation.

Our paper is about a different topic: how to train medical AI (e.g. reasoning language models) to understand of the effects of existing treatments, such as pharmaceuticals. Our position is that popular substitutes for real treatment outcome data (such as clinical guidelines) do not facilitate the development of such AI. The differences between our papers become evident when Joshi et al. (2025) discuss randomized trials. They correctly recognize that conducting RCTs for AI deployments may be infeasibly expensive, and suggest alternative validations. However, in our problem setting, randomized trials are already conducted for the treatments whose effects we wish to understand.

We see no conflict between the perspective of Joshi et al. (2025) and our own; in fact, they are complementary. It would be possible to use their framework to safely deploy the medical AI we advocate for. In the other direction, an AI which precisely understands treatment effects could help manage deployment problems such as the ECG distribution shift mentioned above.

**Causal medical AI**   Many researchers have broadly advocated the application of causal machine learning in medicine (Feuerriegel et al., 2024; Prosperi et al., 2020). Even diagnosis can benefit from some causal reasoning, because of pathophysiology: diseases cause symptoms (Richens et al., 2020). There are multiple causal medical models that use deep learning to predict treatment effects (Zhang et al., 2024; Liu et al., 2024; Nilforoshan et al., 2023). However, there are relatively few language models that attempt to reason about treatment. This paper examines deficiencies in the training and evaluation of models that aim to perform such reasoning.

**Critiques of medical AI evaluation**   The rapid pace of development in medical AI has been met with scrutiny of evaluation standards. Bedi et al. (2024) recently found that only roughly 5% of external, published evaluations of medical AI actually involve real patient data. Raji et al. (2025) decry the use of unrealistic exam questions as benchmarks. Recently, the psychometric notion of construct validity (roughly speaking, the hope that a benchmark actually measures what it says it measures) was used to suggest new benchmarking practices involving observational medical databases (Alaa et al., 2025). This builds upon previous work analyzing the clinical relevance of NLP benchmarks which claim to have biomedical application (Blagec et al., 2023).

**Learning from observational medical databases**   Patient data is both highly private and scattered across the globe. Because of these restrictions, the most successful approaches to learning from large amounts of observational data have involved federated learning (Rieke et al., 2020). These are often organized as collaborative networks among research institutions, such as OHDSI (Hripcsak et al., 2015). They are also established by regulatory agencies for postmarketing surveillance, as in FDA Sentinel (Platt et al., 2018). For evaluating models, MedPerf provides an analogous federated platform (Karargyris et al., 2023).

**Trial emulation as a benchmark**   Some prior works have examined the ability to predict the outcomes of randomized trials in the style of target trial emulation (Hernán & Robins, 2016). Some of these use causal inference methods (specifically, propensity score matching) upon observational data, as a way to assess the capabilities of such algorithms (Forbes & Dahabreh, 2020; Wang et al., 2023). Other works have analyzed the ability to predict future trials from meta-analyses of previous trials (Villar et al., 1995; Cappelleri et al., 1996; LeLorier et al., 1997; DerSimonian & Levine, 1999).

**Health policy metrics**   In public health and health economics, researchers and regulators routinely quantify the cost-effectiveness of improvements of specific policies or improvements to components of the medical system. Holistic metrics include quality-adjusted life years (Feng et al., 2020), event-free life years, mortality indices, and incremental cost effectiveness ratios (Pearson, 2018). Such metrics could be preferable to more formulaic measures, such as mean win rate, for aggregating performance across multiple medical datasets.

## A.2. Experiments with `RxQA` Prompts

### A.2.1. BYPASSING AUTOMATED RxQA CORRECTNESS VERIFICATION

```
Consider the following medication information: {medication_info}
I am using this information to answer the following question: {question}
I believe the correct answer is {answer}. Am I correct?
```

```
Consider the following medication information: {medication_info}
I am using this information to answer the following question: {question}
The correct answer choice was listed as {answer}, and I want to make sure the other
answer choices are incorrect.
Are there multiple correct answers to this question in the answer choices?
```

*Figure 4.* Prompts used by Palepu et al. (2025) for automated verification of the correctness of generated questions and answers within RxQA. Questions which passed these steps, along with an additional step assessing clarity, were sent to pharmacists for manual revision.

```
contemporary FDA product label for KAYDECO (ivacaftor)
```

```
From Figure 1, question about the patient whom Tang et al. (2022) treated successfully with ivacaftor.
```

```
Consider the attached medication information.
I am using this information to answer the attached question.
I believe the correct answer is (D). Am I correct?
```

↓

```
Yes, your assessment that the correct answer is (D) is accurate.  Here's why:

The provided prescribing information for KALYDECO (ivacaftor) explicitly states that
it's indicated for cystic fibrosis (CF) patients with at least one CFTR gene
mutation responsive to ivacaftor.  While the patient has a heterozygous R668C
variant in the CFTR gene, the prescribing information does *not* list R668C as a
mutation responsive to ivacaftor.  Furthermore, the patient's pancreatitis is not
directly related to CF.  Therefore, KALYDECO is not indicated and would be
ineffective in treating the patient's pancreatitis.  Options A, B, and C are
incorrect because there is no indication for use, let alone evidence of
effectiveness.
```

*Figure 5.* Gemini 1.5 Flash falsely verifies the incorrect answer to the question from Figure 1. See Section 3.1 for complete discussion. The correct answer is (C). Gemini 3.5 Flash and Gemini 3.1 Pro Preview also fail in this regard, indicating the problem is not with the model, but rather the verification method based primarily on the FDA product label.

contemporary FDA product label for KAYDECO (ivacaftor)

From Figure 1, question about the patient whom Tang et al. (2022) treated successfully with ivacaftor.

```
Consider the attached medication information.
I am using this information to answer the attached question.
The correct answer choice was listed as (D), and I want to make sure the other answer
choices are incorrect.
Are there multiple correct answers to this question in the answer choices?
```

↓

```
Based on the provided prescribing information for KALYDECO (ivacaftor), the correct
answer is (D) Ineffective.  Here's why the other options are incorrect:

* **(A) Contraindicated for adverse effects:** The prescribing information does *not*
list pancreatitis as a contraindication.  While liver enzyme elevations are a
precaution, this doesn't automatically contraindicate use in all cases.  The
contraindication would need to be directly related to pancreatitis.

* **(B) Effective, with high certainty:** There is no evidence in the provided
information to suggest KALYDECO would be effective in treating pancreatitis.  The
drug targets cystic fibrosis transmembrane conductance regulator (CFTR) mutations,
and while the patient has a CFTR variant, it's of unknown significance and not
directly linked to pancreatitis.  The patient's pancreatitis is likely related to her
methylmalonic acidemia.

* **(C) Effective, with low certainty:**  Similar to (B), there's no basis for even
low certainty of effectiveness.  The connection between the CFTR variant and
pancreatitis is tenuous at best.  The low certainty would imply some plausible
mechanism, which is absent here.

Therefore, only (D) is correct.  There are no multiple correct answers among the
provided options.  The patient's pancreatitis is not related to a CFTR mutation
responsive to ivacaftor.
```

*Figure 6.* Gemini 1.5 Flash falsely eliminates the correct answer (C) to the question from Figure 1. Gemini 3.5 Flash and Gemini 3.1 Pro Preview also fail in this regard.

A.2.2. GENERAL DISCLAIMER ABOUT CASE EXAMPLES

Our counterexamples relating to medication labels and clinical guidelines involve a fair amount of scientific uncertainty. The borderline nature of these instances may seem to undermine the strength of our arguments. It is important to understand our claims, and to recognize our unusually (and asymmetrically) high burden of proof. Benchmarks should generate questions with correct answers, and language models should offer treatment plans that work well. The correctness of these generations can't be checked in the vast majority of situations; we have to rely on the soundness of their underlying methodology. To demonstrate that this methodology is unsound, we have to find examples where there is a substantial amount of published evidence indicating a flaw — yet, this evidence was missed in both the medication label (or guideline), as well as the language model's training. Considering the difficulty in obtaining such instances, we believe our counterexamples are of high quality.

A.2.3. ABLATION: VERIFICATION SUCCEEDS WITH ADDITIONAL INFORMATION

```
# Case
## Patient Profile
- 42-year-old male
- Medical history: Type 2 diabetes mellitus, hypertension, hyperlipidemia
- Presenting with: Chronic calcific pancreatitis
- Clinical severity: M-ANNHEIM IIc

## Clinical Presentation
- 5 hospital admissions over two years for acute pancreatitis
- Strong family history of alcohol-related pancreatitis (father, paternal uncle)
- Heavy alcohol consumption (60-80g daily for >15 years)
- Primary nutrition: Regular diet, poor compliance with low-fat recommendations
- Steatorrhea present (>7g fat/24hr stool collection)

## Diagnostic Findings
- Diffuse pancreatic calcifications on CT scan
- Met American Pancreatic Association criteria for chronic pancreatitis based on
clinical history, imaging, and functional testing
- Average baseline lipase level: 45 U/L (reference range 9-82 U/L)
- Fasting triglycerides: 890 mg/dL (reference range <150 mg/dL)
- Endoscopic ultrasound showed parenchymal changes and ductal dilatation

## Genetic Investigation
- Limited workup for genetic causes of recurrent pancreatitis
- No pathogenic variants in CFTR, PRSS1, or SPINK1 genes identified
- Testing revealed normal CFTR gene sequence without pathogenic variants or variants
of unknown significance

## Additional Workup
- Normal sweat chloride test: 18 mmol/L (reference range <30 mmol/L)
- Pulmonary function testing: Normal, no evidence of obstructive disease
- Fecal elastase-1: 95 µg/g (reference range >200 µg/g), indicating pancreatic
exocrine insufficiency

# Question
Characterize the utility of ivacaftor for resolving this patient's pancreatitis.
(A) Contraindicated for adverse effects.
(B) Effective, with high certainty.
(C) Effective, with low certainty.
(D) Ineffective.
```

*Figure 7.* A synthetic "negative" case of a patient who would not benefit from ivacaftor. This patient's pancreatitis is not idiopathic, as it is likely caused by alcohol use.

contemporary FDA product label for KAYDECO (ivacaftor)     +     published case report of Tang et al. (2022)

From Figure 1, question about the patient whom Tang et al. (2022) treated successfully with ivacaftor.

```
See the attached medical information, as well as the published case study. Answer
the attached question, which is about a different patient than the one in the
published case study.
```

↓

```
Here's an analysis to help answer the question:

[ELIDED FOR BREVITY]

**Answer:**

The most appropriate answer is (C) Effective, with low certainty.

**Reasoning:**

* **Not Contraindicated:** Ivacaftor is not contraindicated for pancreatitis.
* **Low Certainty:** The evidence is based on a single case study.  While the
results are promising, more research is needed to confirm ivacaftor's effectiveness
in treating pancreatitis in CFTR carriers with additional risk factors.
* **Not High Certainty:**  The study doesn't definitively prove causality.  The
patient's improvement could be due to other factors, although the study attempts to
address this.
* **Not Ineffective:** The study shows a positive outcome, suggesting ivacaftor may
have a role, even if not fully understood.

Therefore, while the case study suggests potential benefit, the low number of
patients and the presence of other risk factors make a definitive conclusion
impossible.  More research is needed to establish the effectiveness of ivacaftor for
this specific situation.
```

*Figure 8.* When the case study of Tang et al. (2022) is presented to Gemini 1.5 Flash, it correctly answers (C). This indicates the case study is sufficient evidence to sway its opinion. Gemini 3.5 Flash and Gemini 3.1 Pro Preview answered the same way.

contemporary FDA product label for KAYDECO (ivacaftor)  **+**  published case report of Tang et al. (2022)

From Figure 7, question about patient who would not benefit from ivacaftor.

```
See the attached medical information, as well as the published case study. Answer
the attached question, which is about a different patient than the one in the
published case study.
```

↓

```
The correct answer is (D) Ineffective.

The published case study describes a patient with idiopathic chronic pancreatitis
*and* methylmalonic acidemia who was a CFTR carrier.  Ivacaftor was used successfully
in this specific case because the patient had a CFTR variant that responded to the
drug, and the additional risk factor of methylmalonic acidemia likely contributed to
the pancreatitis.

The case presented in the question is different. This patient has chronic calcific
pancreatitis with a strong family history of alcohol-related pancreatitis and a
history of heavy alcohol use.  The patient also has hypertriglyceridemia and
pancreatic exocrine insufficiency.  There is no indication of a CFTR mutation or
other genetic predisposition that would make ivacaftor a relevant treatment.  The
etiology of the pancreatitis is clearly alcohol-related.

Therefore, ivacaftor would be ineffective for this patient's pancreatitis.  The
underlying cause is alcohol abuse, and the patient's clinical presentation does not
suggest a role for CFTR dysfunction.  Ivacaftor targets CFTR, and this patient does
not have a treatable CFTR-related condition.
```

*Figure 9.* When the negative case from Figure 7 is analyzed in conjunction with the case report of Tang et al. (2022), Gemini 1.5 Flash answers correctly. Gemini 3.5 Flash and Gemini 3.1 Pro Preview answered the same way. This indicates that inclusion of the case report supporting ivacaftor does not somehow overwhelm Gemini's clinical reasoning capability.

### A.2.4. ABLATION: SIMILAR RESULTS FOR COMBINATION THERAPY

In this (somewhat uncertain, but nonetheless informative) ablation, we rule out the possibility that the incorrect verifications observed in Appendix A.2.1 were highly specific to ivacaftor. Trifakta is a combination of three drugs: elexacaftor, tezacaftor, and ivacaftor. It is plausible (though not certain) that it would retain the same effectiveness of plain ivacaftor in treating CFTR-related pancreatitis.

FDA product label for TRIFAKTA (elexacaftor, tezacaftor, and ivacaftor)

Question from Figure 1, with ivacaftor replaced by TRIFAKTA.

```
Consider the attached medication information.
I am using this information to answer the attached question.
The correct answer choice was listed as (D), and I want to make sure the other answer
choices are incorrect.
Are there multiple correct answers to this question in the answer choices?
```

↓

```
Yes, your assessment that the correct answer is (D) is accurate.  Based on the
provided medication guide for Trikafta, and the patient's clinical presentation,
Trikafta is not indicated for this patient's condition.

Here's why:

* **Trikafta's Indication:** Trikafta is specifically indicated for cystic fibrosis
(CF) in patients with certain CFTR gene mutations.  The patient's R668C CFTR variant
is listed in the medication guide as a mutation responsive to Trikafta *in vitro*,
but this does not guarantee clinical benefit, and the patient's pancreatitis is not
related to CF.  Furthermore, the patient declined testing that would confirm the
clinical relevance of this variant.

* **Patient's Condition:** The patient has chronic pancreatitis, a condition
unrelated to CF.  Trikafta does not treat or address the underlying cause of the
patient's pancreatitis.

* **Lack of Evidence:** There's no evidence suggesting Trikafta would be effective
for chronic pancreatitis.

Therefore, Trikafta would be ineffective (D) for this patient's pancreatitis.
Options (A), (B), and (C) are incorrect because there is no contraindication
specifically mentioned for pancreatitis, and there is no evidence to support
effectiveness, high or low certainty.
```

*Figure 10.* Gemini 1.5 Flash verifies a (likely) incorrect answer (D) to a variant of the question from Figure 1.

FDA product label for TRIFAKTA (elexacaftor, tezacaftor, and ivacaftor)

Question from Figure 1, with ivacaftor replaced by TRIFAKTA.

```
Consider the attached medication information.
I am using this information to answer the attached question.
The correct answer choice was listed as (D), and I want to make sure the other answer
choices are incorrect.
Are there multiple correct answers to this question in the answer choices?
```

↓

```
Based on the provided medication guide for TRIKAFTA, there is only one correct answer
to the question regarding its utility for resolving the patient's pancreatitis:

**(D) Ineffective.**

Here's why the other options are incorrect:

* **(A) Contraindicated for adverse effects:**  While TRIKAFTA has a boxed warning
about drug-induced liver injury, there's no contraindication specifically mentioning
pancreatitis.  The medication is indicated for cystic fibrosis (CF) based on
specific CFTR gene mutations.  This patient's pancreatitis is not directly related
to CF.

* **(B) Effective, with high certainty:** There is no evidence or indication in the
provided information that TRIKAFTA would be effective in treating or resolving
chronic pancreatitis.  The patient's CFTR gene variant is of unknown significance,
and the pancreatitis doesn't appear to be related to CF.

* **(C) Effective, with low certainty:**  Similar to (B), there's no basis to
suggest TRIKAFTA would have any effect on the patient's pancreatitis, even with low
certainty.  The lack of a clear link between the R668C CFTR variant and pancreatitis
further weakens this possibility.

Therefore, only (D) accurately reflects the lack of evidence supporting TRIKAFTA's
use in this case.  The medication is for cystic fibrosis, and while the patient has
a CFTR variant, it's of unknown significance and unrelated to her pancreatitis.
There is no indication that TRIKAFTA would be beneficial for her condition.
```

*Figure 11.* Gemini 1.5 Flash invalidates the (likely) correct answer (C) to a variant of the question from Figure 1.

## A.3. Examples Relating to Clinical Guidelines

### A.3.1. PRESCRIBING CASCADE EXAMPLE

An 87-year-old female with a documented medical history of atrial fibrillation, chronic heart failure with preserved ejection fraction (HFpEF), hypertension, hypokalemia, vitamin D deficiency, and glaucoma is taking the following medications:
* Aspirin 81 mg, once daily
* Diltiazem 240 mg extended-release, once daily
* Metoprolol tartrate 50 mg, twice daily
* Furosemide 40 mg, once daily
* Potassium chloride 20 mEq, once daily

The patient is experiencing moderate pedal edema and has an unstable gait. Medication non-adherence is an ongoing issue. Additionally, a recent echocardiogram indicates that the patient's Left Ventricular Ejection Fraction (LVEF) has worsened to less than 40%. What are the next steps of treatment?

↓

Patient Assessment and Current Status
This 87-year-old female presents with worsening heart failure symptoms, including moderate pedal edema, unstable gait, and medication non-adherence. Her recent echocardiogram indicates a transition from heart failure with preserved ejection fraction (HFpEF) to heart failure with reduced ejection fraction (HFrEF), with a left ventricular ejection fraction (LVEF) now <40%. This change necessitates a shift in management strategies to align with guideline-directed medical therapy (GDMT) for HFrEF.[1-3]
Guideline-Directed Medical Therapy (GDMT)
1. Sodium-glucose cotransporter-2 inhibitors (SGLT2i): Initiate dapagliflozin or empagliflozin, as these agents have demonstrated reductions in HF hospitalizations and cardiovascular mortality in both HFrEF and HFpEF populations. SGLT2 inhibitors are administered at fixed doses (e.g., dapagliflozin 10 mg once daily) without titration.[2][4-5]
2. Mineralocorticoid receptor antagonist (MRA): Add spironolactone (starting at 12.5-25 mg daily) or eplerenone, ensuring close monitoring of potassium and renal function to mitigate risks of hyperkalemia and worsening renal function. MRAs are particularly beneficial in HFrEF patients with LVEF <40%.[4-6]
3. Angiotensin receptor-neprilysin inhibitors (ARNIs): Consider sacubitril-valsartan for further optimization of GDMT. ARNIs have shown significant benefits in reducing HF-related morbidity and mortality in HFrEF patients. Transitioning from ACE inhibitors or ARBs to ARNIs requires a washout period of 36 hours to avoid angioedema.[1][7-8]
Symptom Management
1. Loop diuretics: Increase the dose of furosemide to address volume overload and pedal edema. For example, titrate furosemide to 80 mg daily or higher, based on clinical response and monitoring of renal function and electrolytes.[3][9-10]
2. Hypertension management: Optimize antihypertensive therapy using evidence-based agents such as ACE inhibitors, ARBs, or ARNIs, which also provide HF-specific benefits. Beta-blockers (e.g., metoprolol succinate) should be continued for rate control and mortality reduction in HFrEF.[6][8]
Non-Pharmacologic Interventions
1. Heart failure self-care education: Provide detailed counseling on medication adherence, dietary sodium restriction, and daily symptom monitoring (e.g., weight changes, dyspnea). This is critical given the patient's history of non-adherence.[2][9]
2. Exercise and weight-loss programs: Recommend supervised exercise programs to improve functional capacity and quality of life. Weight management may also reduce HF-related symptoms and comorbidities.[2-3]
Multidisciplinary Care
1. Heart failure management program: Refer the patient to a specialized HF clinic for persistent symptoms or recurrent hospitalizations. Multidisciplinary care can improve outcomes through coordinated management.[3][9]
2. Comorbidity evaluation: Assess for sleep apnea, which is common in HF patients and may exacerbate symptoms. Coronary artery disease should also be evaluated, as revascularization may be indicated in select cases.[3][9]
Missing Information
Additional imaging (e.g., cardiac MRI) or biomarkers (e.g., NT-proBNP) may help refine treatment strategies and assess disease severity.

*Figure 12.* Full, unformatted OpenEvidence response for the prompt in Figure 2. Note that the treatment plan is specifically described as "guideline-directed medical therapy."

An 87-year-old female with a documented medical history of atrial fibrillation, chronic heart failure with preserved ejection fraction (HFpEF), hypertension, hypokalemia, vitamin D deficiency, and glaucoma is taking the following medications:
* Aspirin 81 mg, once daily
* Diltiazem 240 mg extended-release, once daily
* Metoprolol tartrate 50 mg, twice daily
* Furosemide 40 mg, once daily
* Potassium chloride 20 mEq, once daily

The patient is experiencing moderate pedal edema and has an unstable gait. Medication non-adherence is an ongoing issue. Additionally, a recent echocardiogram indicates that the patient's Left Ventricular Ejection Fraction (LVEF) has worsened to less than 40%. What are the next steps of treatment?

↓

This is a high-risk medication and HF management question because new LVEF <40% changes the diagnosis and makes current rate-control therapy potentially unsafe. I'll ground the treatment priorities in current HF/AF guidance and label-level medication safety.

[...]

### Assessment

This is no longer HFpEF. LVEF <40% = **HFrEF**, with active congestion and fall risk. The current regimen has two major issues: **diltiazem is problematic in reduced EF**, and **aspirin is not adequate stroke prevention for AF**.

---

### Next Steps

#### 1. Urgent cardiology/HF review
[...]

#### 2. Stop/taper diltiazem and reassess rate/rhythm control
*   Non-dihydropyridine CCBs can worsen HF in impaired LV function; diltiazem plus beta-blocker also increases bradycardia/AV block risk. *(DailyMed)*
*   Consider HFrEF-compatible AF management [...]

#### 3. Convert HF therapy to HFrEF GDMT
*   Start/uptitrate the four foundational classes as tolerated:
    *   ARNI/ACEi/ARB
    *   Evidence-based beta-blocker: metoprolol succinate, carvedilol, or bisoprolol (metoprolol tartrate is not the usual HFrEF formulation).
    *   MRA, if renal function and potassium allow.
    *   SGLT2 inhibitor, if no contraindication.
    *   *Note: HFrEF GDMT includes these four classes, with diuretics for fluid retention.* *(ACC)*

#### 4. Treat congestion
*   Moderate pedal edema suggests inadequate decongestion or nonadherence. Assess weight trend, JVP/lungs, renal function, potassium/magnesium, BP, orthostasis, and adherence before changing loop diuretic intensity.
*   Avoid aggressive unsupervised diuretic escalation in an 87-year-old with gait instability and hypokalemia risk.

#### 5. Anticoagulation decision
[...]

#### 6. Adherence and safety
*   Simplify to once-daily regimens where possible, pharmacy blister packs, caregiver administration, home nursing, BP/weight log, fall-risk/PT assessment, and medication reconciliation.
*   Given unstable gait plus possible anticoagulation, address orthostasis, vision/glaucoma meds, footwear, assistive device, and home hazards.

---

### When to Escalate
[...]

### Citations
[...]

*Figure 13.* Elided response from ChatGPT for Clinicians for the prompt in Figure 2. While it fares slightly better than OpenEvidence by recognizing the importance of medication adherence, it similarly fails to identify the underlying prescribing cascade and does not recommend a reduction in furosemide.

An 87-year-old female with a documented medical history of atrial fibrillation, chronic heart failure with
preserved ejection fraction (HFpEF), hypertension, hypokalemia, vitamin D deficiency, and glaucoma is taking
the following medications:
* Aspirin 81 mg, once daily
* Diltiazem 240 mg extended-release, once daily
* Metoprolol tartrate 50 mg, twice daily
* Furosemide 40 mg, once daily
* Potassium chloride 20 mEq, once daily

The patient is experiencing moderate pedal edema and has an unstable gait. Medication non-adherence is an
ongoing issue. Additionally, a recent echocardiogram indicates that the patient's Left Ventricular Ejection
Fraction (LVEF) has worsened to less than 40%. What are the next steps of treatment?

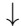

### Assessment
This patient no longer has isolated HFpEF. An LVEF now <40% changes management to heart failure with reduced
ejection fraction (HFrEF) and requires transition to guideline-directed medical therapy (GDMT). Persistent
pedal edema suggests ongoing congestion, and unstable gait plus medication non-adherence substantially
increase fall risk and hospitalization risk.

Several issues in the current regimen are concerning:
* **Diltiazem** is generally not preferred once EF becomes reduced because of negative inotropic effects and
potential worsening heart failure. This concern is especially relevant in patients already taking a beta
blocker.
[...]
* **Medication non-adherence** is a major driver of HF decompensation in older adults.

---

### Recommended next steps

#### 1. Reassess urgently for decompensated HFrEF
[...]

#### 2. Transition from HFpEF regimen to HFrEF GDMT
[...]

#### 3. Address congestion
Moderate pedal edema despite furosemide suggests either:
* non-adherence  * dietary sodium excess  * progressive HF  * inadequate diuresis

**Potential steps:**
* verify adherence  * sodium restriction  * cautious diuretic adjustment  * consider temporary uptitration of
diuretic, with lab monitoring
* *because of her unstable gait, overdiuresis and orthostasis are major risks.*

#### 4. Reassess atrial fibrillation strategy
[...]

#### 5. Address adherence and frailty directly
*Frailty alone is not a reason to withhold GDMT, but titration must be slower and closely monitored. This is
essential.*

**Recommended interventions:**
* pill pack/blister packaging  * home nursing  * caregiver-supervised medication administration  * simplify
regimen  * pharmacy synchronization  * HF disease-management program  * walker/cane evaluation  * PT/fall
risk assessment

---

### When to escalate
[...]

### Citations
[...]

*Figure 14.* Heavily elided response (to fit on the page) from ChatGPT for Healthcare for the prompt in Figure 2. Like ChatGPT
for Clinicians, it addresses medication adherence, but it also fails to detect the prescribing cascade and explicitly suggests temporary
uptitration of the loop diuretic (furosemide) instead of reducing it.

### A.3.2. STALENESS EXAMPLE

A 37-year-old white male with relevant past medicial history of treated HTN, bipolar disorder, and boderline dyslipidmia presented for a physical exam in the ambulatory GIM clinic. He has been on 10 mg of lisinopril and 12.5 mg of hydrochlorothiazide since 2022. He had a coronary calcification testing with a total calcium score of 17 with the circumflex artery (10) and right coronary artery (7) placing him between the 90-100th percentile. His lipid panel was as follows: total cholesterol 197 mg/dL, LDL 126 mg/dL, HDL 62 mg/dL, triglycerides 48 mg/dL, lipoprotein (a) 48 mol/L, and apolipoprotein B 105 mg/dL. The rest of his laboratory work inlcuding complete metabolic profile (CMP), C-reactive protein, and complete blood count were all normal. His blood pressure on the current ambulatory visit was 120/89 and his body mass index was 27.65 kg/m2. What would be the next step in the management of the patient?

$\downarrow$

The next step in the management of this patient should be the initiation of statin therapy. According to the 2018 AHA/ACC/AACVPR/AAPA/ABC/ACPM/ADA/AGS/APhA/ASPC/NLA/PCNA Guideline on the Management of Blood Cholesterol, published by the American College of Cardiology and the American Heart Association, it is reasonable to initiate statin therapy in patients with a coronary artery calcium (CAC) score of 100 or higher or in the 75th percentile or higher[18]. This patient has a CAC score of 17, placing him between the 90-100th percentile, which indicates a higher risk of atherosclerotic cardiovascular disease (ASCVD).

Given his LDL-C level of 126 mg/dL and the presence of risk-enhancing factors such as borderline dyslipidemia and a significant CAC score, moderate- to high-intensity statin therapy is recommended to reduce his LDL-C levels by at least 30-50%[19,20]. The goal is to lower his LDL-C to reduce the risk of future cardiovascular events.

In addition to statin therapy, lifestyle modifications including dietary changes, regular physical activity, and weight management should be reinforced to further reduce cardiovascular risk. Regular follow-up to monitor lipid levels and adherence to therapy is also essential. **Initiate moderate- to high-intensity statin therapy** and reinforce lifestyle modifications as the next steps in management.

*Figure 15.* Hurt et al. (2025) present this answer as an example of OpenEvidence performing effective clinical decision-making. (Original spelling is left intact). Our scrutiny focuses not on the correctness of the final treatment plan, but on part of the underlying clinical reasoning. The answer is based primarily on cheaper, widely-available tests which are the focus of these older guidelines. The answer ignores some of the patient's modern lab tests which, while more costly, are believed to be more accurate measures of atherosclerotic risk. The fundamental problem is that guidelines are not meant to serve as inviolable ground truth about treatment; they are meant to set generic professional standards. As an aside, the clause highlighted in red is not strictly consistent with the guidelines, because the patient is 3 years too young for the relevant guideline condition to apply.

**Background** Atherosclerotic cardiovascular disease is believed to be facilitated by the number of atherogenic lipoprotein particles per liter of serum; see Sniderman et al. (2019) for a more precise understanding. Each such particle contains exactly one apoliprotein B molecule (apoB). LDL-C refers to the mass of cholesterol on LDL particles. Since the amount of cholesterol per particle may vary, among both particles and persons, LDL-C is a rough proxy of apoB. Furthermore, LDL-C is not actually measured, but estimated from other values via the Friedewald equation. Many researchers support apoB's use as a primary measure (Mortensen, 2024; Sniderman et al., 2024; De Oliveira-Gomes et al., 2024; Cole et al., 2023; Contois et al., 2023). The National Lipid Association (Soffer et al., 2024) issued a statement supporting this role. apoB a co-primary metric in the 2021 CCA guidelines (Pearson et al., 2021). Even the 2018 AHA guidelines themselves acknowledge the superiority of apoB in subpopulations (Grundy et al., 2019). However, LDL-C can be measured by a cheap, widely-available enzymatic assay. apoB requires an immunoassay which, in 2018, may have been prohibitive at a population scale. Furthermore, many trials of statins were conducted with LDL-C endpoints.

### A.4. Analysis of HealthBench Professional

We classified the 1135 criteria in HealthBench Professional as follows.

| Status | Count | Meaning |
|---|---|---|
| **Grounded** | 883 | Verified support by a verbatim passage in the authoritative guideline. |
| **Grounded (Unverified)** | 7 | Supporting passage found but final verification remains pending. |
| **Not Grounded (Absent)** | 65 | Referenced guideline contains no information regarding the criterion. |
| **Not Grounded (Disagree)** | 15 | Guideline contains related information that contradicts the criterion. |
| **Not Clinical** | 134 | Relates to non-medical aspects such as formatting or context seeking. |
| **Absence of Evidence** | 14 | Guideline explicitly identifies a lack of consensus or evidence gap. |
| **Partial/Incorrect Grounding** | 17 | Extraction attempted but found insufficient during verification. |

$883/(1135-134) \approx 88\%$ of the criteria are classified as grounded. This claim depends on the correct filtering of non-clinical criteria as well as the correct grounding of clinical criteria. To filter non-clinical criteria, we used a language model (Gemini 3 Flash Preview) and manually verified the 134 negative cases. To make grounding determinations, an agentic workflow was followed using Gemini 3 Flash Preview. Following manual inspection and verification, we presume a 2% error rate (at most) and conservatively claim that 86% of the criteria are grounded. Further details on this process, including criteria-level grounding results, are in the repository accompanying this paper.

### A.5. Systematic Literature Analysis Methodology and Results

We used Gemini 3 Flash Preview to classify the 789 nominally treatment-related studies from Chen et al. (2026). For each study, the model was provided with the title and abstract and asked to classify the following aspects:

- Treatment Focus: The evaluation was classified as treatment-focused if the majority of LLM queries measured an understanding of the causal consequences of a treatment action, such as recommending a therapy or estimating a treatment effect. It was classified as not treatment-focused if the evaluation was restricted to diagnostic accuracy, prognostic risk, or other intermediate predictive tasks, even if treatment-related materials or scenarios were involved.

- Benchmarking Reusability: The evaluation was classified as reusable if it was structured as an automated, mechanically reproducible benchmark consisting of standardized patient cases paired with explicit, pre-defined reference labels. It was classified as not reusable if it relied on one-time, subjective clinician or expert grading of LLM outputs without a reference standard.

- Nature of Ground Truth: The reference standard used to establish the correctness of the LLM's decisions was classified into one of six categories: standard clinical guidelines, subjective clinician or expert opinion, product prescribing labels, real patient treatment outcomes in observational data (such as electronic health records or clinical registries), real treatment outcomes in randomized controlled clinical trials, or other and unsure.

The classifications were manually adjudicated or verified. The complete categorization of benchmarking reusability and treatment focus across the 789 studies is presented in the following table.

| | Not Treatment Focused | Treatment Focused | Total |
|---|---|---|---|
| **Not Reusable** | 111 | 398 | 509 |
| **Reusable** | 100 | 180 | 280 |
| **Total** | **211** | **578** | **789** |

Additionally, the following table provides the complete categorical breakdown of the ground truth standards across both the entire treatment-related corpus and the subset of treatment-focused, reusable benchmarks.

| Ground Truth Category | All Studies | Reusable Treatment Benchmarks |
|---|---|---|
| Guidelines | 303 | 98 |
| Human Opinion | 399 | 58 |
| Real Treatment Outcomes (Observational) | 47 | 16 |
| Real Treatment Outcomes (Randomized Trial) | 4 | 0 |
| Product Labels | 4 | 0 |
| Other | 31 | 7 |
| Unsure | 1 | 1 |
| **Total** | **789** | **180** |

Study-level classifications are provided in the repository accompanying this paper.

