# OpenReview forum: "Position: Medical AI Neglects Real Treatment Outcomes"
_ICML.cc/2026/Position_Paper_Track — ICML 2026 Position Paper Track regular_

### Official Review · Reviewer_XRFP · 2026-03-08

**Significance:** 4
**Argument Clarity:** 3
**Rating:** 5
**Confidence:** 4

**Questions:**

Please refer to strengths and weaknesses.

**Alternative Views Section:**

Yes

**Compliance With Llm Reviewing Policy A Conservative:**

Affirmed.

**Discussion Potential:**

4

**Final Justification:**

I kept my score after seeing the rebuttal.

**Paper Summary:**

The position of this paper is that medical AI should be really trained and evaluated on the long-term patient outcomes. The paper used several examples to support the position.

**Position:**

Yes

**Position In Title:**

Yes

**Related Work:**

3

**Strengths And Weaknesses:**

This is a well-written and argued position paper. The paper points out a critical point that is missing in today's medical AI, that is, the lack of evaluation of an AI product against the long-term patient outcomes.
The strength of the position paper is to reveal and substantiate an overlooked aspect of the medical AI, that is, what real evaluation of an AI product should be. Patient outcomes were demonstrated to play a critical role in determining the success of medical AI.

The position that "the long-term goal of medical AI should be to help write clinical practice guidelines and other authoritative
syntheses, rather than to merely read and reference them." is not well supported, and the meaning of this position is not clear. Does it mean that the authors suggest that AI should overturn the established clinical guidelines? If so, what are the benefits and risks it will bring up? The authors should elaborate on this.

**Support:**

4

---

> ### Author Rebuttal · Authors · 2026-03-29
>
> We appreciate your positive review. Thank you for your support.
>
> We mentioned that long-term goal in order to clearly distinguish the medical AI we envision from the medical AI of today. (It is clear that today's AI cannot help write guidelines, because it is trained to cite them). To clarify, the long-term goal would be to improve, not overturn, clinical guidelines. We promote scalable, comprehensive training approaches and rigorous evaluation methodology so that AI can be (eventually) be a trusted, supportive technical tool for writing guidelines, much like systematic reviews and meta-analyses are used today. Potential benefits would be:
> * fast updates: guidelines currently take years to write and are often severely out of date
> * precision: rather than condensing practice to 100 pages, guidelines could expand to accommodate more special cases
> * comprehensiveness: guidelines could capture the accumulated clinical experience of millions of practitioners on billions of patients, rather than just a handful of expert opinions and publications
> * accuracy: language models are more nuanced than basic statistical methods like meta-analysis.
>
> Some potential downsides:
> * the AI may not properly account for noise or other inadequacies in its training data
> * the evaluations of the AI may not sufficiently expose its potential flaws. (These could arise from an overly reductive formalization of treatment understanding)
>
> Please let us know if this answers your question. We are happy to elaborate further.
>
> If an evidently superintelligent medical AI is developed using real treatment outcomes, there will robust debates about whether it actually achieves that status, and about what its role should be. Our paper argues that, insofar as medical treatment is concerned, we are not currently on the path to building AI worthy of such debate.

---

> > ### Author Rebuttal · Reviewer_XRFP · 2026-04-02
> >
> > I thank the authors for the rebuttal.

---

### Official Review · Reviewer_wJcM · 2026-03-08

**Significance:** 3
**Argument Clarity:** 3
**Rating:** 5
**Confidence:** 3

**Questions:**

Are there specific existing datasets that currently approximate the proposed "treatment outcome benchmarks"?

How can privacy-preserving techniques realistically scale to support the massive data requirements of outcome-based foundation model training?

**Alternative Views Section:**

Yes

**Compliance With Llm Reviewing Policy A Conservative:**

Affirmed.

**Discussion Potential:**

3

**Final Justification:**

My concerns are resolved with the paper and rebuttal.

**Paper Summary:**

This position paper argues that medical AI research has fundamentally drifted away from the ultimate goal of medicine: improving patient treatment outcomes. The authors contend that current models (especially Medical LLMs) are over-optimized for "intermediate" tasks—such as diagnosis, knowledge recall, and clinical guideline adherence—because they are primarily trained on textual syntheses (publications, textbooks) rather than real-world longitudinal data reflecting actual treatment interventions and their results. The paper supports this claim by analyzing training corpora of major medical foundation models and critiquing popular benchmarks that rely on physician opinion rather than causal treatment effects.

**Position:**

Yes

**Position In Title:**

Yes

**Related Work:**

3

**Strengths And Weaknesses:**

Strengths

- Challenges the prevailing methodology of "medical knowledge" models by reframing the objective around patient outcomes.

- Uses a diverse range of evidence, including training data analysis, benchmark critiques, and specific clinical case studies.

- Addresses a critical gap in the safety and utility of AI systems deployed in real-world decision support.

- Provides a clear research agenda for outcome-based training and evaluation rather than just pointing out flaws.

Weaknesses

- The paper does not fully resolve the massive practical hurdles of training foundation models on longitudinal data, such as privacy, data standardization, and global scaling.

- The critique of current benchmarks is largely grounded in discussion and specific examples rather than a large-scale quantitative empirical analysis.

- The argument may overlook existing specialized research in causal inference and clinical trial prediction that does focus on outcomes.

**Support:**

3

---

> ### Author Rebuttal · Authors · 2026-03-29
>
> Thank you for your supportive and encouraging review.
>
> **W1. Longitudinal Data:** In Section 5, we will add more detail about how modern data pipelines and permission schemes are easing the challenges of accessing and working with longitudinal data. An entire "real world evidence" industry rapidly extracts causal insights from large-scale longitudinal data, but these techniques are not used in medical AI training. When it comes to the practical hurdles of integrating such methods: *where there is a will, there is a way*. It's important to circulate the message of this paper in order to generate demand for new training pipelines. If we don't acknowledge the problems with text corpora, then medical AI won't migrate away from them.
>
> **W2. Prevalence:** We will mention that Bedi et al. found that just 9% of medical AI evaluations nominally evaluate treatment understanding [1]. Our fine-grained analysis improves upon that rough estimate in two key ways. First, we show those nominal claims are weak: they often do not focus on, or satisfactorily evaluate, understanding of treatment. Further, we assess MedHELM, a major evaluation which claims to be holistic in of itself.
>
> To further quantify the prevalence of the problems we have identified, we have added a new, systematic analysis of the literature. We begin with the recent review of [2], which found 789 out of 4609 clinical evaluations of LLMs (17.1%) nominally involve treatment planning or recommendation. We further scrutinize these 789 studies, checking (1) if they actually focused on some aspect of treatment itself (rather than diagnosis, prognosis, or other intermediate tasks), (2) whether they could be reused as benchmarks (as opposed to being one-time manual assessments of answers), and (3) whether real treatment outcomes were used as ground truth (as opposed to syntheses or opinions). We find:
> * Just 180 (3.9%) are potentially-reusable benchmarks focused on treatment, and
> * The vast majority (702 / 789 = 88.9%) of ground truth is based on guidelines or human opinion. Only 51 involve real treatment outcomes. That drops to just 16 when considering only the 180 potentially-reusable benchmarks focused on treatment.
>
> This systematic analysis supports our claims that (1) even nominally treatment-related evaluations do not focus on such assessment, and (2) the use of textual syntheses or opinions as ground truth is pervasive.
>
> **W3. Specialized Models:** Appendix A.1 mentions this gap. Frontier medical LLMs are now routinely used to answer questions about medical treatment, yet causal ML techniques play no immediate role in their development, and are instead relegated to specialized research models. We'd like to see that change.
>
> **Q1. Closest Approximations:** The closest approximations are the trial outcome prediction benchmarks discussed in Section 4.2. It should be possible to repurpose such benchmarks to focus on patient treatment outcomes rather than trial success. (As Section 4.2 explains, these sound quite similar, but involve significant methodological distinctions).
>
> **Q2. Scaling Privacy:** This is an interesting and important question. One possible approach is to exploit structure in the learning problem. Unlike general differentially-private learning, treatment effect inferences are about larger (unseen) populations, not just about the training dataset, and certainly not just about individuals within that set. This could enable restrictions of the query interface and/or alternative formulations of privacy [3]. Another approach is to use federated learning and exploit the gap between intra-institution and inter-institution privacy requirements. Until such technologies are developed, users of sensitive medical AI could be bound by data use agreements, as is standard practice in the medical field.
>
> [1] Bedi, Suhana, et al. "Testing and evaluation of health care applications of large language models: a systematic review." Jama 333.4 (2025): 319-328.
>
> [2] Chen, Sully F., et al. "LLM-assisted systematic review of large language models in clinical medicine." Nature medicine (2026): 1-8.
>
> [3] Bhaskar, Raghav, et al. "Noiseless database privacy." International Conference on the Theory and Application of Cryptology and Information Security. Berlin, Heidelberg: Springer Berlin Heidelberg, 2011.

---

> > ### Author Rebuttal · Reviewer_wJcM · 2026-04-01
> >
> > The authors' responses have addressed most of my concerns. I will keep my positive recommendation for this paper.

---

### Official Review · Reviewer_reze · 2026-03-13

**Significance:** 2
**Argument Clarity:** 2
**Rating:** 4
**Confidence:** 4

**Questions:**

How would your proposed training-and-evaluation framework handle medical AI tasks whose objective is not directly tied to longitudinal patient outcomes (image segmentation, medical history summarization, clinical documentation, or information extraction) without unfairly treating those areas as lower-value or methodologically deficient?

**Alternative Views Section:**

Yes

**Compliance With Llm Reviewing Policy A Conservative:**

Affirmed.

**Discussion Potential:**

1

**Final Justification:**

Thanks for your rebuttal. The clarification for the intermediate tasks (their value and connection to longitudinal output) fully resolved my concern. I have increased my rating accordingly.

**Paper Summary:**

This position paper argues the current medical AI are optimized for the wrong targets. The paper highlights that existing foundation models rely heavily on static "ground truth" sources, for example, medication labels and physician consensus, which often fail to capture complex clinical scenarios. The paper then argues that intermediate predictive accuracy does not automatically guarantee better health in the future, therefore, the paper proposes that medical AI must be trained on longitudinal patient records and evaluated using clinical trials.

**Position:**

Yes

**Position In Title:**

Yes

**Related Work:**

2

**Strengths And Weaknesses:**

## Strengths
1. The position is healthy to the community. It forces the community to recognize that benchmark performance is often a poor proxy for real clinical value.
2. The paper is well-written, with many concrete examples to illustrate and motivate the position.

## Weaknesses
1. My biggest concern for the paper is treating "real treatment outcomes" as the only gold standard for all medical AI. It ignores that medical AI is not a single task family. While disease diagnosis and prognosis are very important areas in machine learning for health, image segmentation, risk stratification, medical history summarization, and etc are also very important. In those areas, the link between the tasks' objectives and the longitudinal outcomes may be completely indirect or nonexistent.
2. This is related to the first weakness, but by aggressively judging the entire field against a single, outcome-centered metric, the paper implicitly downgrades large, highly valuable sectors of medical AI research. This makes the paper feel more like a goalkeeper than a carefully scoped scientific position that may leads to more insightful discussion.
3. The proposed solution leans heavily on shifting training toward longitudinal records (like EHRs and claims data), but it severely glosses over reality. This data is notoriously messy, highly confounded, unevenly available, and often a terrible fit for the abstraction levels required by tasks like image segmentation or clinical summarization. The position should discuss what can researchers do to resolve this questions, especially for sub areas where the link to the longitudinal outcomes is not obvious.

**Support:**

2

---

> ### Author Rebuttal · Authors · 2026-03-29
>
> We thank you for relaying your concerns, to which we are very receptive. We are committed to expressing our position in a way that is instructive and inclusive to everyone in medical AI. In short: we aim to expand the scope of medical AI research by using better data about treatment. We do not broadly discourage other work; in fact, we anticipate synergy among these directions. Please let us know if the following changes are satisfactory.
>
> **W2. The Value of Intermediate Tasks:** We will curtail language which potentially suggests intermediate tasks are inherently less valuable. There is the harsh possibility that *some* intermediate tasks do not contribute much to patient outcomes; we stand by that fact. However, we can enunciate this concern while avoiding "guilt by association". Our discussion will emphasize specific mechanisms which lead to problems (e.g. side effects, different incentives, misaligned losses) rather than intermediate-ness per se. To further soften our tone, we will explicitly acknowledge that many intermediate tasks have demonstrably high ROI. Finally, in the abstract, we will clarify that improving treatment outcomes is the *eventual* goal of all medical AI.
>
> By any fair accounting, understanding of treatment should be improved. Highlighting this gap is not equivalent to devaluing other tasks. We regret if any language suggested otherwise.
>
> **W1. Connecting Intermediate Tasks to Outcomes:** In Section 5, we will elaborate how work on "intermediate" tasks will thrive in an outcome-oriented research agenda. For example, consider ambient AI summarization of patient conversations - a valuable intermediate task. These algorithms are benchmarked on NLP-style metrics such as WER or MTR, which is fine. We encourage an *additional* (not replacement) layer of research tying these metrics to clinician performance [1] and, in turn, improved patient experience [2]. This burden would not fall on algorithmic researchers who apply their techniques to medicine; it would be the purview of researchers working on evaluation and deployment.
>
> Even this additional layer does not need to completely, explicitly link intermediate tasks to treatment outcomes. This pattern is familiar to ML/AI practitioners: the relationship between pretraining objectives and end goals isn't fully elucidated. Yet, recognizing the utility of pretraining does not contradict the primacy of end-goal performance.
>
> In ambient AI, anticipating likely treatments can help the doctor ask for vital information to complete their note. Of course, having a good note helps select the right treatment. Highlighting that different tasks service the same ultimate goal (outcomes) enables fruitful collaboration rather than antagonism.
>
> **W3. Longitudinal Data:** In Section 5, we will add more detail about how modern data pipelines and permission schemes are easing the challenges of accessing and working with longitudinal data. An entire "real world evidence" industry rapidly extracts causal insights from large-scale longitudinal data, but these techniques are not used in medical AI training. When it comes to the practical hurdles of integrating such methods: *where there is a will, there is a way*. It's important to circulate the message of this paper in order to generate demand for new training pipelines. If we don't acknowledge the problems with text corpora, then medical AI won't migrate away from them.
>
> Longitudinal data need not be incorporated into every medical ML task. As discussed in Appendix A.1, a novel AI which understands treatment effects could assist the development and deployment of AI for intermediate tasks. Researchers working on intermediate tasks could use such a model to estimate downstream effects rather than having to work with real treatment outcomes themselves.
>
> **4. Summary:** We will add a segment to Section 6 about your overall concern about intermediate tasks, incorporating the clarifications above. To answer **your question** about proxy/intermediate objectives: these should be handled the same way the medical research community handles them. Empirically, the shift towards evidence-based, patient-centered outcomes did not imperil the in-silico or in-vitro biomedical investigation of molecules and analytes. We do not anticipate that training medical AI to better understand treatment will devalue or degrade its performance on existing medical AI tasks.
>
> **Conclusion**. These changes are just minor updates and are consistent with our position. Again, we thank you for helping refine our arguments.
>
> [1] Moramarco, Francesco, et al. "Human evaluation and correlation with automatic metrics in consultation note generation." Proceedings of the 60th Annual Meeting of the Association for Computational Linguistics (Volume 1: Long Papers). 2022.
>
> [2] Afshar, Majid, et al. "A pragmatic randomized controlled trial of ambient artificial intelligence to improve health practitioner well-being." NEJM AI 2.12 (2025): AIoa2500945.

---

> > ### Author Rebuttal · Reviewer_reze · 2026-04-02
> >
> > Thanks for your rebuttal. The clarification for the intermediate tasks (their value and connection to longitudinal output) fully resolved my concern. I have increased my rating accordingly.

---

### Official Review · Reviewer_JAqZ · 2026-03-13

**Significance:** 4
**Argument Clarity:** 4
**Rating:** 5
**Confidence:** 3

**Questions:**

None

**Alternative Views Section:**

Yes

**Compliance With Llm Reviewing Policy A Conservative:**

Affirmed.

**Discussion Potential:**

4

**Final Justification:**

My concerns are resolved with the paper and rebuttal.

**Paper Summary:**

This paper argues that medical AI neglects real treatment outcomes in both training and evaluation. Models are trained primarily on publications, clinical guidelines, and drug labels rather than actual patient records, and evaluated using physician opinion and guideline-based benchmarks that have fundamental limitations. The authors call for incorporating longitudinal patient data into training, using randomized experiments for evaluation, and reestablishing the improvement of real treatment outcomes as the central goal of medical AI.

**Position:**

Yes

**Position In Title:**

Yes

**Related Work:**

4

**Strengths And Weaknesses:**

## Strengths

- The paper presents **multiple layers of evidence**: a systematic literature survey (Table 1), author-conducted experiments, and a quantitative re-analysis of publicly available HealthBench data, rather than relying on a single line of reasoning.
- The **ivacaftor experiment is methodologically careful** — it tests multiple models (Gemini 1.5 Flash, 2.0 Flash, Gemma 3 27B) and includes ablation studies.
The HealthBench analysis is concise but compelling, using OpenAI's own data to demonstrate that "physician-level agreement" provides only a weak correctness guarantee.
- The argument covers **three distinct levels of failure** (training data, evaluation metrics, and benchmark design), giving it both breadth and depth.

## Weaknesses

- The paper does not **quantify the prevalence of the identified problems**. Two case studies demonstrate that failures exist, but do not establish how widespread they are — for instance, what proportion of RxQA questions involve off-label drug use, or how many answers were incorrectly verified.
- **The leap from individual cases to broad conclusions is occasionally too large**. The prescribing cascade example reflects the behavior of one specific system (OpenEvidence), yet the conclusion is generalized to guideline-reliant medical AI as a whole.
The recommendations remain underspecified.

**Support:**

4

---

> ### Author Rebuttal · Authors · 2026-03-29
>
> We appreciate your positive assessment and feel encouraged by it. Thank you.
>
> **W1. Prevalence:** To quantify the prevalence of the problems we have identified, we have added a new, systematic analysis of the literature. We begin with the recent review of [1], which found 789 out of 4609 clinical evaluations of LLMs (17.1%) nominally involve treatment planning or recommendation. We further scrutinize these 789 studies, checking (1) if they actually focused on some aspect of treatment itself (rather than diagnosis, prognosis, or other intermediate tasks), (2) whether they could be reused as benchmarks (as opposed to being one-time manual assessments of answers), and (3) whether real treatment outcomes were used as ground truth (as opposed to syntheses or opinions). We find:
> * Just 180 / 4609 = 3.9% are potentially-reusable benchmarks focused on treatment, and
> * The vast majority (702 / 789 = 88.9%) of ground truth is based on guidelines or human opinion. Only 51 involve real treatment outcomes. That drops to just 16 when considering only the 180 potentially-reusable benchmarks focused on treatment.
>
> This systematic analysis supports our claims that (1) even nominally treatment-related evaluations do not focus on such assessment, and (2) the use of textual syntheses or opinions as ground truth is pervasive. For more discussion on this topic, see our response below to Reviewer wJcM below.
>
> Quantifying some other aspects (e.g. how often would RxQA/TreatmentPC mislabel questions) is not possible due to fundamental scientific limitations. To show RxQA/TreatmentPC would mislabel the ivacaftor question, we had to be fortunate enough to both (1) find an incorrect labeling, and (2, the hard part) find a compelling body of scientific evidence which refutes the labeling. As noted in Appendix A.2.2, it is difficult to mount successful arguments from this rhetorically disadvantaged position; we are glad you found ours compelling. In practice, clinical guidelines can have shortcomings, but most often these just remain informally recognized by practitioners until a new consensus forms after the long, expensive cycle of formal study and scientific publication.
>
> **W2. Generalization:** For completeness, we have added the responses of the other models besides OpenEvidence; they make the same mistake. This is intuitive, because our argument applies to any guideline-constrained response. We highlighted OpenEvidence simply because it explicitly denotes, within its responses, its adherence to particular guidelines.
>
> **Side note:** we will publish all of GPT 4's answers to the meta-evaluation, which HealthBench used and analyzed but did not disclose. (This will save future researchers some costs).
>
> [1] Chen, Sully F., et al. "LLM-assisted systematic review of large language models in clinical medicine." Nature medicine (2026): 1-8.

---

> > ### Author Rebuttal · Reviewer_JAqZ · 2026-04-02
> >
> > Thanks for the rebuttal. My concerns are resolved.

---

### Decision · Program_Chairs · 2026-04-30

**Decision:**

Accept (regular)

**Comment:**

Reviewers agreed that this position targets a meaningful topic, medical AI, and highlights that current evaluation practices drawn from the general domain are not appropriate in context of this application area. This is particularly concerning in medicine where an overemphasis on intermediate task performance can create an illusion of progress without delivering meaningful improvements. Reviewers also agree that the range of evidence strengthens the foundation of this position. In particular, the analyses of HealthBench is compelling since it was presented as evidence of improvement and still fails to provide a sufficiently strong correctness guarantee.

There are several concerns raised during review, that are worth consideration. There are recurrent concerns regarding the incorporation of longitudinal patient data. Specifically, longitudinal outcomes reflect only a subset of potential value in medical AI and there are numerous challenges with respect to privacy, quality and scalability that are not easy to navigate. Further, there is a potential gap between "real treatment outcomes" and "best treatment outcomes" for many patients based on exogenous factors such as insurance, availability of care, and historical biases --- a potential alternative view in favor of different outcomes. Another consideration that could similarly merit an alternative view might be whether this is already know to the clinical subcommunity. Finally, while not discussed thoroughly in review there does seem to be an emerging literature on the use of longitudinal data:

- Renc, P., Jia, Y., Samir, A.E. et al. Zero shot health trajectory prediction using transformer. npj Digit. Med. 7, 256 (2024). https://doi.org/10.1038/s41746-024-01235-0
- Waxler, S., Blazek, P., White, D., et al. Generative medical event models improve with scale. arXiv preprint (2025). arXiv:2508.12104.
- Shmatko, A., Jung, A.W., Gaurav, K. et al. Learning the natural history of human disease with generative transformers. Nature 647, 248–256 (2025). https://doi.org/10.1038/s41586-025-09529-3

These are, of course, fundamentally different types of models but all focused on incorporating longitudinal outcomes and so it would be useful to understand the relationship between this position and these lines of work or explicitly relegate them out of scope for consideration.

Overall, reviewer interest suggests there may be interest among the broader audience of ICML, though this interest may be limited if these are already acknowledged concerns of the clinical subcommunity.